# Horse Placental Extract Enhances Neurogenesis in the Presence of Amyloid β

**DOI:** 10.3390/nu13051672

**Published:** 2021-05-14

**Authors:** Andreia de Toledo, Kaori Nomoto, Eiichi Hirano, Chihiro Tohda

**Affiliations:** 1Research Institute Japan Bio Products Co., Ltd. Kurume 839-0864, Japan; a_toledo@placenta-jbp.co.jp (A.d.T.); kanomoto@inm.u-toyama.ac.jp (K.N.); ehirano@placenta-jbp.co.jp (E.H.); 2Section of Neuromedical Science, Division of Bioscience, Institute of Natural Medicine, University of Toyama, Toyama 930-0194, Japan

**Keywords:** horse placental extract, neurogenesis, dendrite, Alzheimer’s disease, memory recovery

## Abstract

Human placental extract and animal-derived placental extracts from pigs and horses host a wide range of biological activities. Several placental products are used as medicines, cosmetics, and healthcare substances worldwide. However, the use of placental extracts for neuronal functioning is currently not established because the number of relevant studies is limited. A few previous reports suggested the neuroprotective effect and dendrite genesis effect of placental extract. However, no studies have reported on neurogenesis in placental extracts. Therefore, we aimed to investigate the effects of horse placental extract on neurogenesis, and we examined the protective effect of the extract on the onset of memory disorder. A horse placental extract, JBP-F-02, was used in this study. JBP-F-02 treatment dose-dependently increased the number of neural stem cells and dendrite length under Aβ treatment in primary cultured cortical cells. The oral administration of JBP-F-02 to a 5XFAD mouse model of Alzheimer’s disease at a young age significantly prevented the onset of memory dysfunction. This study suggests that the extract has the potential to prevent dementia.

## 1. Introduction

Human placental extract and animal-derived placental extracts from pigs and horses host a wide range of biological activities. Several placental products are used as medicines, cosmetics, and healthcare substances worldwide. However, the use of placental extract for neuronal functioning has not yet been established because the number of relevant studies is limited. In chronically stressed and ovariectomized mice, treatment with porcine placental extract for 5 weeks protected against memory decline and neuronal death [1]. Our previous study indicated that the administration of porcine placental extract for 4 weeks facilitated memory functioning in aged (76 weeks old) mice [2]. We also previously reported that the administration of either human placental extract or porcine placental extract for 15 days significantly improved memory disorder in a 5XFAD mouse model of post-onset Alzheimer’s disease (AD) [3]. In the 5XFAD mouse, intracellular Aβ starts to accumulate in 1.5 month old mice, while the extracellular deposition is present in mice as young as 2 months [4]. Hyperphosphorylation of tau is also escalated in broad areas of the brain [5]. Disruption of axons [5] and dendritic loss [6] are also observed at least in the hippocampus and prefrontal cortex. In the brains of 5XFAD mice treated with human placental extract, a decline in dendrite density was recovered [3]. An increase in dendrite length was also achieved by human placental extract in amyloid β (Aβ)-treated primary cultured cortical neurons [3]. These data suggest that dendrite growth is an important factor in placental extract-induced memory recovery. However, neurogenesis is also an important factor in maintaining cognitive functioning. Although a decline of neurogenesis in the hippocampal dentate gyrus occurs at 12–14 weeks of age in 5XFAD mice [7,8], no studies have reported on neurogenesis in placental extracts. Adult neurogenesis has received attention in the AD field as a possible preventative and therapeutic strategy. Therefore, we aimed to investigate the effects of horse placental extract on neurogenesis. The protective effect of the extract on the onset of memory disorder was also examined. 

## 2. Materials and Methods

### 2.1. Materials

The horse placental extract, JBP-F-02, was manufactured by Japan Bio Products (Tokyo, Japan). The active partial fragment of Aβ, Aβ25–35 (Sigma-Aldrich, St. Louis, MO, USA), was dissolved in sterile distilled water and incubated for 7 days at 37 °C once before use. Full-length Aβ1–42 was dissolved in dimethyl sulfoxide (DMSO) at 5 mM and diluted in Ham’s F-12 medium at 100 μM, followed by incubation for 24 h at 4 °C once before use. The supernatant obtained after centrifugation for 10 min at 4 °C was used as aggregated Aβ1–42. 

### 2.2. Animals

Transgenic mice (5XFAD) were obtained from The Jackson Laboratory (Bar Harbor, ME, USA). The 5XFAD mice have five mutations: Swedish (K670N and M671L), Florida (I716V), and London (V717I) in human APP695 cDNA and human PS1 cDNA (M146L and L286V) under the transcriptional control of the neuron-specific mouse Thy-1 promoter 6 [4]. They were obtained by crossing hemizygous transgenic mice with B6/SJL F1 breeders. To investigate the effect of JBP-F-02 on 5XFAD, we used pre-onset hemizygous 5XFAD mice (male and female, 13 weeks old) and non-transgenic littermate wildtype mice (male and female, 13 weeks old), which were obtained by crossing a hemizygous 5XFAD mouse and a B6/SJL F1 mouse. All mice were housed with free access to food and water and were maintained in a controlled environment (22 ± 2 °C, 50% ± 5% humidity, 12 h light/dark cycle starting at 7:00 a.m.). 

JBP-F-02 was mixed with food powder (Labo MR Stock, Nosan Corporation, Yokohama, Japan) at concentrations of 0.03% and 3% and then solidified. The average daily feeding amount was 5 g in all groups. Wildtype mice were administered normal feed (*n* = 6). 5XFAD mice were divided to three groups and administered normal feed (*n* = 6), 0.03% JBP-F-02 (*n* = 4), or 3% JBP-F-02 (*n* = 5). Feeding duration was 58 days.

### 2.3. Object Recognition Memory Test

The object recognition memory test was performed 43 days after the oral administration of JBP-F-02 (Figure 1). Two identical objects (colored ceramic ornaments) were placed at a fixed distance within a square box (30 cm × 40 cm; height, 36.5 cm, 60–84 lx). A mouse was then placed at the center of the box, and the number of contacts with the two objects was recorded during a 10 min period (training session). Twenty-four hours after, the mice were placed back into the same box. In the box, one of the objects was replaced with a novel object (another ceramic ornament with a different shape and color). The number of contacts with the two objects was recorded during a 10 min period (test session). A preference index was used to measure the cognitive functions for objects. The index was defined as the ratio of the number of contacts with any of the objects (training session) or the novel object (test session) divided by the total number of contacts with both objects. 

### 2.4. Locomotive Function in Open-Field Test

On feeding day 52, each mouse was gently released in the center of a black box and allowed to move freely for 10 min. A black open box (49 cm × 49 cm; height, 48.5 cm) was used for the locomotion test. The locus of the mouse in the box was recorded using a digital camera. The total travel distance and the time spent in the peripheral zone in 10 min were analyzed using EthoVision 3.0 (Noldus, Wageningen, The Netherlands). The peripheral zone was defined as a 5 cm wide area from the edge line.

### 2.5. Western Blotting

On feeding day 58, the mouse brain cortex was homogenized with M-PER (Thermo Fisher Scientific, Waltham, MA, USA) containing 1 × Halt protease and phosphatase inhibitor cocktail (Thermo Fisher Scientific). Samples were mixed with NuPAGE lithium dodecyl sulfate sample buffer (Thermo Fisher Scientific) containing 5% 2-mercaptoethanol (Wako, Osaka, Japan) at 75 °C for 5 min and loaded onto a 14% sodium dodecyl sulfate polyacrylamide gel (SDS–PAGE) (50 µg/lane). After electrophoresis, proteins in the gel were transferred to a nitrocellulose membrane (Bio-Rad, Hercules, CA, USA). After blocking with 0.1% T-TBS containing 5% skim milk (Wako) at room temperature, the membrane was incubated with a mouse monoclonal anti-Aβ antibody (1:1000, clone 6E10, BioLegend, San Diego, CA, USA) or mouse monoclonal anti-GAPDH (1:10000, clone 5A12, Wako) in Can Get Signal solution 1 (Toyobo, Osaka, Japan) overnight at 4 °C. After washing with 0.1% T-TBS, the membrane was incubated with a horseradish peroxidase-conjugated secondary antibody against mouse IgG (1:2000; Cat. No. sc-2005, Santa Cruz) in Can Get Signal Solution 2 (Toyobo) for 2 h at room temperature. After washing, chemiluminescence on the membrane was detected by ECL Prime Western Blotting Detection Reagent (GE Healthcare, Chicago, IL, USA) using an ImageQuant LAS 4000 system (GE Healthcare).

### 2.6. Primary Culture

The primary culture was prepared as previously described [3,5]. Embryos were removed from the ddY mother mouse (Japan SLC, Shizuoka, Japan) at 14 days of gestation. The cortices were dissected, and the dura mater was removed. The tissues were minced, dissociated by a 0.05% trypsin-EDTA solution (Thermo Fisher Scientific), and grown in neurobasal medium (Thermo Fisher Scientific) containing 12% B-27 supplement (Thermo Fisher Scientific), 0.6% d-glucose, and 2 mM l-glutamine in eight-well chamber slides (Falcon, Franklin Lakes, NJ, USA) coated with 5 μg/mL poly-d-lysine at 37 °C in a humidified incubator with 10% CO_2_. The seeding cell density was 4.4 × 10^4^ cells/cm^2^. 

### 2.7. Measurement of Dendritic Density and Number of Cells

To measure the density of dendrites, after 3 days of nontreatment incubation, the cells were treated with or without 10 μM Aβ25–35 or 1 μM Aβ1–42 and simultaneously treated with JBP-F-02 (0.02–2 mg/mL) or vehicle solution (distilled water) for another 4 days. As described previously [3], the neurons were fixed with 4% paraformaldehyde for 60 min and immunostained with a polyclonal antibody against microtubule-associated protein 2 (MAP2, 1:2000, Abcam, Cambridge, UK) as a dendritic marker. Alexa Fluor 488-conjugated goat anti-rabbit IgG (1:600) was used as the secondary antibody (Molecular Probes, Eugene, OR, USA). Nuclear counterstaining was performed using 4’,6-diamidino-2-phenylindole dihydrochloride (DAPI, 1 μg/mL, Sigma-Aldrich). Fluorescence images were captured with a 10× objective lens using a fluorescence microscope system (Cell Observer, Carl Zeiss, Tokyo, Japan). Ninety-eight, 32, and 32 images were captured per treatment in Figure 2 Figure 3 Figure 4, respectively. The lengths of the MAP2-positive dendrites were measured using a MetaMorph analyzer (Molecular Devices, Sunnyvale, CA, USA), which automatically traced and measured the neurite lengths without measuring the cell bodies. The sum of the dendrite lengths was divided by the number of MAP2-positive neurons. The number of MAP2-positive cells and MAP2-negative cells was independently counted using MetaMorph. For nestin staining, a monoclonal antibody against nestin (1:100, Chemicon, Temecula, CA, USA) and a polyclonal antibody against MAP2 (1:2000, Abcam) were used. Alexa Fluor 488-conjugated goat anti-mouse IgG (1:600) and Alexa Fluor 594-conjugated goat anti-rabbit IgG (1:600) were used as secondary antibodies (Molecular Probes). Double-positive cells to nestin and MAP2 were detected as nestin-positive neurons using MetaMorph. 

### 2.8. Statistical Analysis

Data are expressed as the mean ± 95% confidence interval (CI). To determine the statistically significant differences, GraphPad Prism 6 (GraphPad Software, San Diego, CA, USA) was used. Since the data in Figure 1 passed the normality test, parametric tests were performed. A two-tailed paired *t*-test (Figure 1A), one-way analysis of variance (ANOVA), post hoc Bonferroni’s multiple comparisons test (Figure 1B,C,E), repeated-measures two-way ANOVA, and post hoc Bonferroni’s multiple comparisons test (Figure 1D) were applied. Data in Figure 2 Figure 3 Figure 4 respectively, revealed mean values and their distribution in similarly treated cells, but these were not related to individual differences, which are generally considered parametric data. Therefore, one-way ANOVA with a post hoc Bonferroni’s multiple comparisons test was performed. The effect size (*r*) was calculated and shown in graphs. The significance level was set at *p* < 0.05. The criteria for small, medium, and large effect sizes in the *t*-test were 0.2, 0.5, and 0.8, respectively. The criteria for small, medium, and large effect sizes in one-way ANOVA were 0.1, 0.3, and 0.5, respectively.

**Figure 1 nutrients-13-01672-f001:**
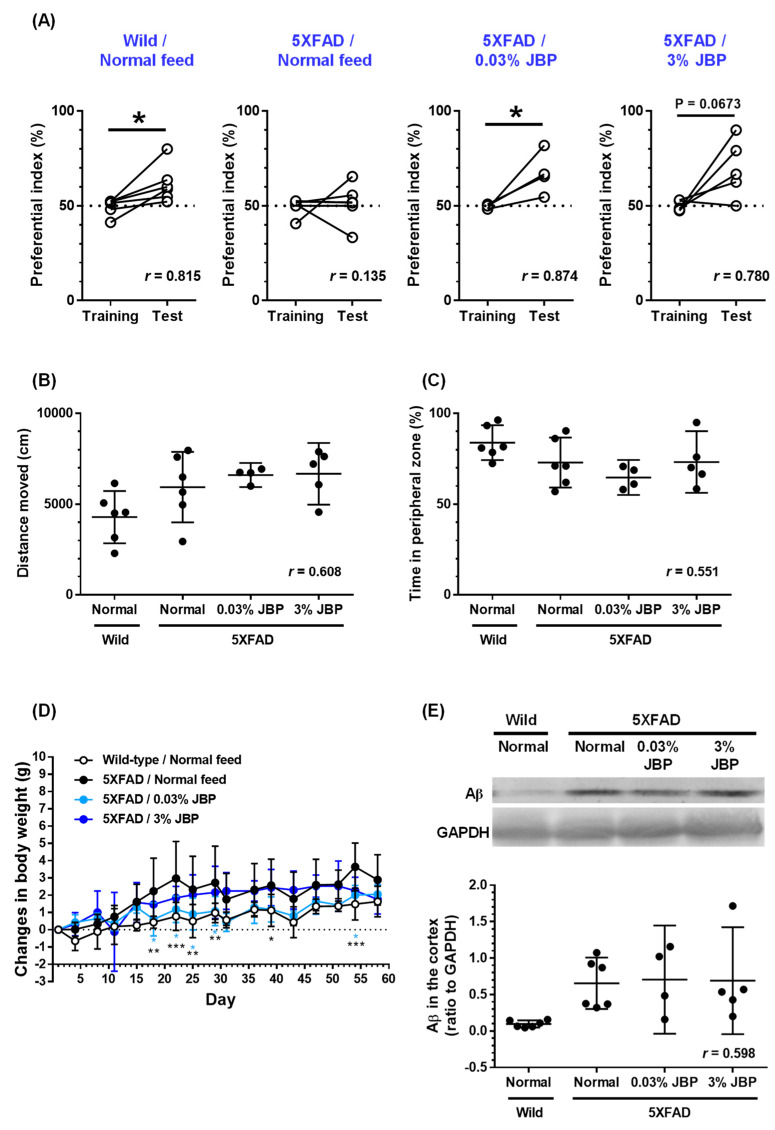
Effect of oral administration of JBP-F-02 on object recognition memory deficit in young 5XFAD mice. (**A**) JBP-F-02 mixed feed (0.03% and 3%) or a normal feed was administered for 58 days to mice (male and female, 13 weeks old). An object recognition test was carried out at day 43 with a 24 h interval between the training session and test session. The preferential indices of the training and test sessions are shown; * *p* < 0.05, two-tailed paired *t*-test. (**B**) Distance moved in an open field for 10 min; *p* > 0.05, one-way ANOVA. (**C**) Spent time as a percentage in peripheral zone of an open field. The peripheral zone was defined as a 5 cm wide area from the edge line; *p* > 0.05, one-way ANOVA. (**D**) Changes in body weights of mice during the experimental period; * *p* < 0.05, ** *p* < 0.01, *** *p* < 0.001 vs. normal feeding 5XFAD mice, repeated measures two-way ANOVA with post hoc Bonferroni’s test. (**E**) Oligomeric Aβ expression in the cerebral cortex; one-way ANOVA, *post hoc* Bonferroni’s test. (**A**–**E**) *n* = 4–6 mice. Means *±* 95% CI. ANOVA, analysis of variance.

## 3. Results

### 3.1. JBP-F-02 Prevented Onset of Memory Dysfunction in 5XFAD Mice 

Our previous study indicated the memory recovery effect of human placental extract product and porcine placental extract in an aged 5XFAD mouse model of AD [3]. However, no studies have reported a preventive effect of placental extract on AD model mice in the pre-onset period. Since 5XFAD mice exhibit memory deficits 16–20 weeks after birth [4,9,10], we used younger 5XFAD mice at 13 weeks of age. Previous studies of porcine placental extract showed that doses of 242 and 2424 mg/kg/day extract were administered via gavage to 5XFAD mice [3]. Another study on porcine placental extract in aged mice indicated that the effective doses of the extract were 1000 and 5000 mg/kg/day [2]. Therefore, we set the feeding doses for the present study as 0.03% and 3%, which are almost equal to 50 mg/kg/day and 5000 mg/kg/day, respectively, because the average daily feeding amount was 5 g.

Administration of the horse placental extract product JBP-F-02 was totally continued for 58 days. Object recognition memory was evaluated with a 24 h interval at day 43. In the training session, all four groups showed an approximately 50% preference index. In the test session, wildtype mice administered normal feed showed significantly high exploratory behavior toward the novel object, indicating normally retained memory (Figure 1A). In contrast, the exploratory behavior toward the novel object in the normal feeding 5XFAD mice was close to chance (50%), indicating memory dysfunction. Feed intake with 0.03% JBP-F-02 significantly improved the ability of object recognition memory. A high dose of JBP-F-02 (3%) showed an almost significant upregulation of object recognition memory. The effect sizes of 0.03% and 3% JBP-F-02 intakes were sufficiently large. After the memory test, the locomotive function was evaluated using an open-field test. Distances moved in 10 min did not differ among the groups (Figure 1B). We also evaluated the staying time in the peripheral zone in the open-field test because it reflects the anxiety level in mice. No differences were observed between the groups (Figure 1C). Changes in body weight were plotted over the entire period from the start of feeding (Figure 1D). Repeated-measures two-way ANOVA indicated a significant two-factor (feeding × day) interaction in normal feeding wildtype mice and normal feeding 5XFAD mice (F (16,160) = 1.825, *p* = 0.032). Between normal feeding 5XFAD mice and 0.03% JBP-F-02 fed 5XFAD mice, a significant feeding × day interaction was observed (F (16,128) = 1.904, *p* = 0.0255). Post hoc Bonferroni’s multiple comparison test indicated significant differences in changes in body weight at 18, 22, 25, 29, and 54 days after intake between normal feeding 5XFAD mice and 0.03% JBP-F-02 feeding 5XFAD mice.

Overproduction of oligomeric Aβ begins at 2 months of age and continues to increase in 5XFAD mice [4]. Oligomeric Aβ expression in the cerebral cortex was quantified by Western blotting. Oligomeric Aβ was detected as a band of 10–20 kDa. Aβ expression in the cortex was normalized to that of GAPDH. Normal feeding 5XFAD mice showed high expression of oligomeric Aβ compared with wildtype mice (Figure 1E). JBP-F-02 feeding at 0.03% and 3% did not change the amount of oligomeric Aβ.

### 3.2. JBP-F-02 Protects against Decline in Neuron Numbers under Aβ Treatment

In our previous in vitro study, an Aβ25–35 solution was prepared by incubation for 3 days at 37 °C for aggregation [3]. In the condition, Aβ25–35 induced weak neuronal death, but remarkably decreased dendrite density. To increase Aβ25–35 cell death toxicity, the incubation duration should be extended [11]. In this study, we prepared Aβ25–35 using a 7 day incubation for the neuronal toxicity assay using primary cultured cortical neurons. As a result, cells treated with Aβ25–35 for 4 days significantly decreased the number of neurons (Figure 2B). The simultaneous addition of JBP-F-02 protected against neuronal loss in a dose-dependent manner. MAP2-positive cells were specified as neurons and MAP2-negative cells were counted as non-neurons. The number of non-neurons in Aβ25–35- and vehicle solution-treated cells increased, and the 0.4 and 2 mg/mL JBP-F-02 treatments significantly reduced the number of non-neurons (Figure 2C). Dendrite length was lowered by Aβ25–35 treatment, and the 0.2, 0.4, and 2 mg/mL JBP-F-02 treatments significantly increased dendrite density (Figure 2D,E). The effect sizes of JBP-F-02 treatment in Figure 2B,D were sufficiently large.

If the increase in neuron numbers by JBP-F-02 (Figure 2B) is affected by neurogenesis, this would denote that the neurogenesis phenomenon occurred early on, and that the newborn cells would mature to neurons. Therefore, we evaluated neurogenesis within 48 h following JBP-F-02 treatment with effective doses (0.2–2 mg/mL). Nestin is a neural stem/progenitor cell marker. We evaluated the number of nestin-positive cells after Aβ25–35 treatment and the related effects of JBP-F-02. Aβ25–35 treatment itself did not change the number of nestin-positive cells from 6 to 48 h. At 24 h and 48 h after JBP-F-02 treatment, nestin-positive cells significantly increased at all doses (0.2, 0.4, and 2 mg/mL) compared with vehicle-treated cells (Figure 3B,C). An increase in the number of neurons by JBP-F-02 was also confirmed under full-length Aβ1–42 treatment. Aβ1–42 did not reduce neuron numbers after 4 days of treatment because 1 μM Aβ1–42 induced only neurite atrophy without cell death. Under these conditions, JBP-F-02 treatment increased nestin-positive neuronal numbers (Figure 4B). The effect sizes of JBP-F-02 treatment in Figure 3B and Figure 4B were sufficiently large. These results suggest that neurogenesis might be enhanced by JBP-F-02 under Aβ treatment.

## 4. Discussion

A previous report on porcine placental extract in chronically stressed and ovariectomized mice demonstrated protection against a decline in neuronal number in the hippocampal CA3 region [1]. Another study using porcine placental extract showed no change in neuronal numbers in the hippocampus after extract treatment in aged mice, although memory deficits were recovered [2]. These previous reports suggest a neuroprotective effect of the placental extract. However, the increase in neurogenesis by placental extract treatment has not yet been reported or discussed. This study showed that horse placental extract, JBP-F-02, increased the number of neural stem cells and dendrite length under Aβ treatment in primary cultured cortical cells. JBP-F-02 also prevented the onset of memory dysfunction in the 5XFAD mice. As shown in Figure 3; Figure 4, Aβ treatment itself did not decrease nestin-positive neurons, indicating that the reduced number of neurons by Aβ25–35 (Figure 2B) was due to cell death. Although it has not been addressed whether JBP-F-02 protects against cell death, we found that JBP-F-02 increased the number of newborn neurons. Although the placental extract itself contains diverse chemicals and amino acids, specific active constituents have not been identified in any case. JBP-F-02 contains multiple constituents that are not annotated. Inputs of neurogenesis regulation are generally considered to be many kinds of neurotransmitters, such as GABA, glutamate, serotonin, norepinephrine, and acetylcholine [12]. Adult neurogenesis linage is pushed forward via regulating several key transcriptional factors such as Pax6, Tbr2, NeuroD, and Tbr1 [13]. At least glutamate is contained in human placenta extract [14]. The signaling mechanism of JBP-F-02 for neurogenesis will be investigated upon the identification of active constituents contributing to neurogenesis and related transcriptional factors. The identification of active constituents for neurogenesis in JBP-F-02 is currently under investigation. Concretely, we try to identify active constituents which are transferred into the brain after oral administration of JBP-F-02 using LC–MS/MS annotation.

Ramirez et al. reported that Aβ25–35 treatment decreases dendrite length, spine density, and neurogenesis in the hippocampus, and Aβ25–35 treatment induces memory dysfunction [15]. In 5XFAD mice, neuronal death occurs later than 12 months of age and in a very limited area, i.e., cortical layer V [16]. In contrast, a decline in neurogenesis is observed at a young age of 8 weeks in 5XFAD mice [17]. Therefore, enhancing neurogenesis might be effective in protecting cognitive impairment at an early stage.

This study has some limitations as it is unknown which constituents of JBP-F-02 are related to phenotypes, how this stimulates neurogenesis, and whether the mechanisms of neurogenesis and dendrite genesis overlap or are independent. As a dendrite genesis mechanism, we focus on δ-catenin as one of key molecules, as δ-catenin localizes to dendrites and dendritic spines in mature neurons [18] and is required for dendritic formation in the cerebral cortex [19]. Regulation of δ-catenin by JBP-F-02 will be investigated in a future study.

The results obtained in the present study suggest that the effects of JBP-F-02 on memory maintenance are likely mediated by neurogenesis and dendrite genesis. In our previous study, treatment with human placental extract significantly increased dendrite density in the cerebral cortex and memory function in aged 5XFAD mice [3]. The supply of newborn neurons might lead to the consequent upregulation of dendrite density. We need to reveal the causal relationship between neurogenesis and dendrite genesis in the future.

## 5. Conclusions

This study showed for the first time that horse placental extract, JBP-F-02, prevented the onset of memory dysfunction in 5XFAD mice and enhanced neurogenesis and dendrite density under Aβ treatment. The extract has potential as a preventive medicine for dementia.

## Figures and Tables

**Figure 2 nutrients-13-01672-f002:**
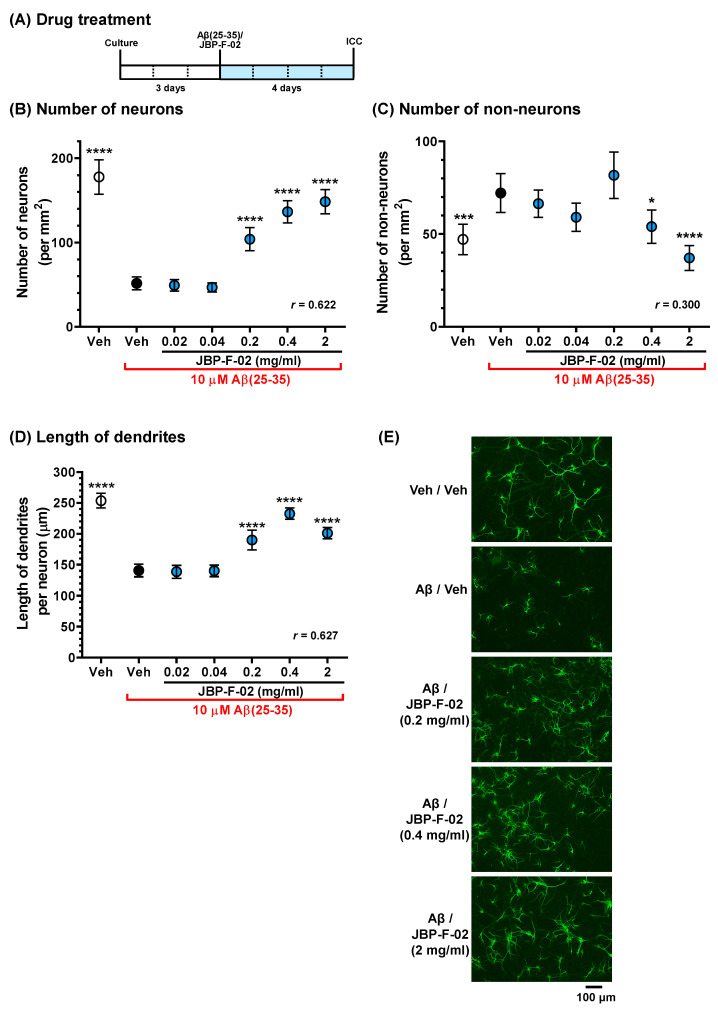
Effects of JBP-F-02 on Aβ-induced neuronal loss and dendrite atrophy. (**A**) Cortical neurons were cultured for 3 days and then treated with or without aggregated Aβ25–35 (10 μM). At the time of Aβ25–35 addition, the cells were simultaneously treated with JBP-F-02 at 0.02, 0.04, 0.2, 0.4, and 2 mg/mL concentration or a vehicle solution (distilled water). Four days after the treatment, the cells were fixed and immunostained for MAP2. Counter staining was done by DAPI. (**B**) MAP2-positive and DAPI-positive cells were counted as neurons. (**C**) MAP2-negative and DAPI-positive cells were counted as non-neurons. (**D**) The length of MAP2-positive dendrites. (**E**) Representative photos of the immunostained dendrites are shown; * *p* < 0.05, *** *p* < 0.001, **** *p* < 0.0001 vs. Aβ25–35-treated and vehicle solution-treated cells (black circle), one-way analysis of variance with post hoc Bonferroni’s test, *n* = 98 photos. Scale bar indicates 100 μm. Means *±* 95% CI. Aβ, amyloid β; DAPI, 4’,6-diamidino-2-phenylindole dihydrochloride.

**Figure 3 nutrients-13-01672-f003:**
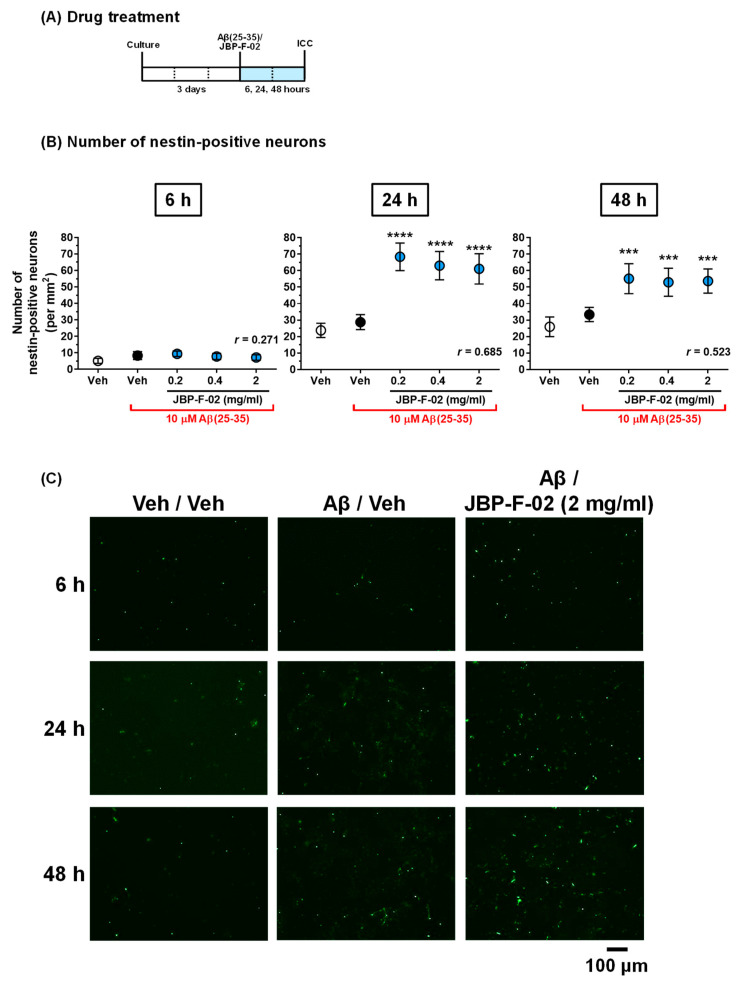
Effect of JBP-F-02 on neurogenesis under Aβ25–35 treatment. (**A**) Cortical neurons were cultured for 3 days and then treated with or without aggregated Aβ25–35 (10 μM). At the time of Aβ25–35 addition, the cells were simultaneously treated with JBP-F-02 at 0.2, 0.4, and 2 mg/mL concentration or a vehicle solution (distilled water). Six, 24, and 48 h after the treatment, the cells were fixed and immunostained for nestin and MAP2. (**B**) Nestin-positive and MAP2-positive cells were counted as neuronal stem cells. (**C**) Representative photos of the immunostained nestin are shown; *** *p* < 0.001, **** *p* < 0.0001 vs. Aβ25–35-treated and vehicle solution-treated dells (black columns), one-way analysis of variance with post hoc Bonferroni’s test, *n* = 32 photos. Scale bar indicates 100 μm. Means *±* 95% CI. Aβ, amyloid β.

**Figure 4 nutrients-13-01672-f004:**
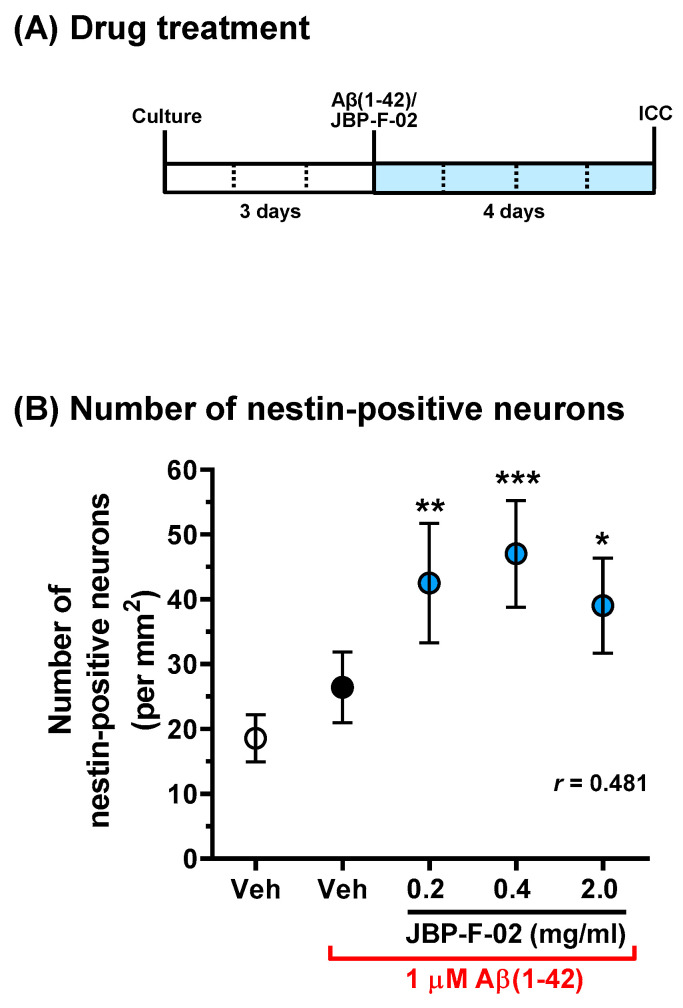
Effect of JBP-F-02 on neurogenesis under Aβ1–42 treatment. (**A**) Cortical neurons were cultured for 3 days and then treated with or without aggregated Aβ1–42 (1 μM). At the time of Aβ1–42 addition, the cells were simultaneously treated with JBP-F-02 at 0.2, 0.4, and 2 mg/mL concentration or a vehicle solution (distilled water). Four days after the treatment, the cells were fixed and immunostained for nestin and MAP2. (**B**) Nestin-positive and MAP2-positive cells were counted as neuronal stem cells; * *p* < 0.05, ** *p* < 0.01, *** *p* < 0.001 vs. Aβ1–42-treated and vehicle solution-treated cells (black circle), one-way analysis of variance with post hoc Bonferroni’s test, *n* = 32 photos. Means *±* 95% CI. Aβ, amyloid β.

## Data Availability

The data presented in this study are available upon request from the corresponding author.

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
