# Peer review of "Horse Placental Extract Enhances Neurogenesis in the Presence of Amyloid β"

_nutrients, 2021, doi:10.3390/nu13051672_

Round 1

Reviewer 1 Report

The manuscript entitled “Horse placental extract enhances neurogenesis in the presence  of amyloid β” describes the neuroprotective/neurogenic effect of JBP-F-02  in “in vivo” and “in vitro”  experiments. The results obtained by the authors are interesting, the manuscript is informative, however  there are also some shortcomings:

 The “Material and method” should be more detailed:

  1. How long the experimental animals were treatment with the JBD-F-02 ( this information is noted in the “Results” sections, but it should be mentioned firstly in” Material and methods”)
  2. The authors should describe experimental groups (in "in vivo" studies) , and dosage of JBD-F-02 (in "in vitro" experiments).
  3. How many animals were used in each experimental groups?
  4. The authors should mentioned that the locomotion functions were measured by Open field test. It is indicated only in the “Results” section.
  5. When the animal tissues were collected? Just after behavioral tests?
  6. The nestin staining description should be more detailed (time of incubation, secondary antibodies, etc)

The “Results” section:

  1. Line 169: could the authors explain what does the “drug × test interaction” mean?
  2. Line 198: the authors wrote: “we prepared Aβ25-35 via a 7-day incubation for neuronal toxicity assay using primary cultured cortical neurons in this study. Treatment with Aβ25-35 for 4 days significantly decreased the number of neurons”. The incubation time of the primary neurons was 7 or 4 days? The diagram (Fig 2A) points that it lasted 4 days.
  3. Fig 2E. Is the description of the left bottom photomicrograph appropriate? There is indicated as JBP-F-02 (0.2 mg/ml), but according the graphs 2B and 2D density of the neurons/ length of dendrites at this solution  (0.2mg/ml) is quite similar like at the 2 mg/ml. The photomitographs did not reflect   
  4. Why the experiment evaluated the number of nestin- positive cells  has the parameters other than the previous experiments (time points,  dosage of JBP-F-02)? Could the authors explain it?
  5. Fig 4B: the legend should be changed on “ The number of nestin-positive neurons”. The present version “the number of neurons” is not appropriate.

 The “Discussion” section is the weakest part of the manuscript. It is too vague.  I recommend to improve it.

Author Response

  1. How long the experimental animals were treatment with the JBD-F-02 ( this information is noted in the “Results” sections, but it should be mentioned firstly in” Material and methods”)
    Reply: We added information in Materials and Methods (L.77).

  2. The authors should describe experimental groups (in "in vivo" studies) , and dosage of JBD-F-02 (in "in vitro" experiments).
    Reply: We added information in Materials and Methods (L.75-77, L.137).

  3. How many animals were used in each experimental groups?
    Reply: We added information in Materials and Methods (L.75-77).

  4. The authors should mentioned that the locomotion functions were measured by Open field test. It is indicated only in the “Results” section.

Reply: We revised expressions and added detail information in 2.4. (L.96-102)

  1. When the animal tissues were collected? Just after behavioral tests?
    Reply: We mentioned the day of sacrifice (L.105).

  2. The nestin staining description should be more detailed (time of incubation, secondary antibodies, etc)
    Reply: More detailed protocol was mentioned for nestin staining (L.151-156).

The “Results” section:

  1. Line 169: could the authors explain what does the “drug × test interaction” mean?
    Reply: We revised correctly this part as “significant 2 factors (feeding x day)” (L.202, L.205).

  2. Line 198: the authors wrote: “we prepared Aβ25-35 via a 7-day incubation for neuronal toxicity assay using primary cultured cortical neurons in this study. Treatment with Aβ25-35 for 4 days significantly decreased the number of neurons”. The incubation time of the primary neurons was 7 or 4 days? The diagram (Fig 2A) points that it lasted 4 days.

Reply: To avoid confusing, we added “once before use” at L.55 and L.57. In addition, we revised the explanation as follows (L.229-L.235); “In our previous in vitro study, Aβ25-35 solution was prepared by an incubation for 3 days at 37 °C for aggregation [3]. In the condition, Aβ25-35 induced weak neuronal death, but remarkably decreased dendrite density. To increase Aβ25-35 cell death toxicity, an incubation duration should be extended [11]. In this study, we prepared Aβ25-35 via a 7-day incubation for neuronal toxicity assay using primary cultured cortical neurons. As a result, treatment cells with Aβ25-35 for 4 days significantly decreased the number of neurons (Figure 2B). “

  1. Fig 2E. Is the description of the left bottom photomicrograph appropriate? There is indicated as JBP-F-02 (0.2 mg/ml), but according the graphs 2B and 2D density of the neurons/ length of dendrites at this solution  (0.2mg/ml) is quite similar like at the 2 mg/ml. The photomitographs did not reflect   
    Reply: More representative photos were replaced in Figure 2E.

  2. Why the experiment evaluated the number of nestin- positive cells  has the parameters other than the previous experiments (time points,  dosage of JBP-F-02)? Could the authors explain it?
    Reply: We explained about dosage and treatment duration of JBP-F-02 in nestin staining as follows (L.255-L.258); “If the increase in neuron numbers by JBP-F-02 (Figure 2B) would be affected by neurogenesis, the neurogenesis phenomenon occurred early, and then the new born cells would maturate to neurons. Therefore, we evaluated neurogenesis within 48 h by JBP-F-02 treatment with effective doses (0.2 – 2 mg/ml).”

  3. Fig 4B: the legend should be changed on “ The number of nestin-positive neurons”. The present version “the number of neurons” is not appropriate.
    Reply: All figure legends were corrected.

 The “Discussion” section is the weakest part of the manuscript. It is too vague.  I recommend to improve it.

Reply: We newly mentioned in Discussion about analyzing neurogenesis mechanism and dendrite genesis mechanism as follows with new references (L.313-L.320; L.327-L.332); “Adult neurogenesis linage is pushed forward via regulating several key transcriptional factors such as Pax6, Tbr2, NeuroD and Tbr1 [13]. At least glutamate is contained in human placenta extract [14]. Signaling mechanism of JBP-F-02 for neurogenesis will be investigated by identification of active constituents contributing neurogenesis and re-lated transcriptional factors. The identification of active constituents for neurogenesis in JBP-F-02 is currently under investigation. Concretely, we try to identify active con-stituents which are transferred in the brain after oral administration of JBP-F-02 by LC-MS/MS annotation.” “This study has some limitations as it is unknown which constituents of JBP-F-02 are related to phenotypes, how this stimulates neurogenesis, or whether the mecha-nisms of neurogenesis and dendrite genesis overlap or are independent. As a dendrite genesis mechanism, we focus on the δ-catenin pathway as one of the candidates. δ-catenin localizes to dendrites and dendritic spines in mature neurons [18] and is re-quired for dendritic formation in the cerebral cortex [19].”

Reviewer 2 Report

Nutrients

Ms. Ref: nutrients-1178615

Title: “Horse placental extract enhances neurogenesis in the presence of amyloid β”

General comment

The authors investigated whether the injection of horse placental extract may induce a sort of brain-resilience effect to incipient accumulation of Aβ through putative neurogenesis-mediated pathways. The authors leverage their experience in such a study design and methodological approach and reach the evidence that the drug investigated shows a dose-dependent increase in the number of neural stem cells and dendrite length under Aβ monomers exposure in primary cultured cortical cells.

This exciting study explores a potential therapeutic avenue for Alzheimer’s disease biology to prevent severe cognitive decline and clinical progression of the disease. Although most of the manuscript is well-written and the pharmacological rationale is adequately fleshed out, there are some points in the introduction, methodology and protocol description, and Discussion that fails to get the message across and, in some cases, sounds controversial and are not entirely convincing. In addition, the soundness of the narrative needs some improvement to avoid any misunderstanding on the study design or the clinical-pharmacological conclusions drawn.

Even the statistical Workplan necessitates editing work to let out its strengths and potential caveats as required by standard journal requirements.

Such shortcomings hinder the manuscript publication suitability and should be addressed to consider the piece sufficiently developed for moving forward.

Please, find below a few comments.

Major points:

1. The authors employed a well-characterized mouse model of AD, the 5XFAD. This model expresses express human APP and PSEN1 transgenes and higher levels of amyloidogenic by-products and cerebral accumulation of Aβ species, besides other features of AD endophenotypes. However, the authors do not adequately introduce the model itself and its underlying biology and neuropathology (see, for instance Sadleir KR et al. Mol Neurodegener. 2015 Jan 7;10:1. doi: 10.1186/1750-1326-10-1.)

Moreover, the authors introduce the model as Aβ-treated subjects, which is somewhat confusing and may yield some concerns about the methodological approach. This issue turns up also in the Discussion where sentences such as “The consequent neuronal number under Aβ treatment is the sum of surviving neurons and newborn neurons..” are challenging to interpret.

A more structured introduction of other studies in this direction, apart from the authors’ previous research work, may enrich the overall clinical value of the findings here reported.

2. The authors relied too heavily on p-values and did not provide other critical statistic information that would help elaborate on the clinical meaningfulness of the study. Confidence intervals, effect size coefficient and balancing data interpretation in light of effect size measures over the p-value. Otherwise, the authors should explain why they deem there is no risk of an over-interpretation of the significance of p-values. There is no mention in the statistical section regarding whether comparison tests have been corrected for multiple comparisons to avoid type I error. Finally, it is not clear how whether the authors assessed normality distribution tests before rolling out the comparison and correlation analysis workflow.

3. The authors state:”This study has some limitations as it is unknown which constituents of JBP-F-02 are related to phenotypes, how this stimulates neurogenesis, or whether the mechanisms of neurogenesis and dendrite genesis overlap or are independent.” However, they do not provide any forward-looking argumentation about potential tolls, studies, and solutions to overcome these methodological and conceptual barriers. This standpoint would support replication studies and increase scientific attention towards this study and its potential.

Minor points

The author state, “The primary culture was prepared as previously described”, “The dendrite length was evaluated as described previously”,

I suggest providing supplementary material and specifying whether any change to the previous protocol was made or not.

Some expressions, such as ”previous reports” are overused. I suggest improving the language with a wider variety in the wording/phrasing.

Author Response

Comments and Answers

We thank the editors and reviewers for their positive comments, thorough evaluation of our manuscript, and for the opportunity to improve our paper. We provide here a point-by-point response to the reviewers' comments that we believe thoroughly address their concerns. The revised parts are shown in yellow in the main text.

Reply to Referee 2

Ms. Ref: nutrients-1178615

Title: “Horse placental extract enhances neurogenesis in the presence of amyloid β”

General comment

The authors investigated whether the injection of horse placental extract may induce a sort of brain-resilience effect to incipient accumulation of Aβ through putative neurogenesis-mediated pathways. The authors leverage their experience in such a study design and methodological approach and reach the evidence that the drug investigated shows a dose-dependent increase in the number of neural stem cells and dendrite length under Aβ monomers exposure in primary cultured cortical cells.

This exciting study explores a potential therapeutic avenue for Alzheimer’s disease biology to prevent severe cognitive decline and clinical progression of the disease. Although most of the manuscript is well-written and the pharmacological rationale is adequately fleshed out, there are some points in the introduction, methodology and protocol description, and Discussion that fails to get the message across and, in some cases, sounds controversial and are not entirely convincing. In addition, the soundness of the narrative needs some improvement to avoid any misunderstanding on the study design or the clinical-pharmacological conclusions drawn.

Even the statistical Workplan necessitates editing work to let out its strengths and potential caveats as required by standard journal requirements.

Such shortcomings hinder the manuscript publication suitability and should be addressed to consider the piece sufficiently developed for moving forward.

Please, find below a few comments.

Major points:

  1. The authors employed a well-characterized mouse model of AD, the 5XFAD. This model expresses express human APP and PSEN1 transgenes and higher levels of amyloidogenic by-products and cerebral accumulation of Aβ species, besides other features of AD endophenotypes. However, the authors do not adequately introduce the model itself and its underlying biology and neuropathology (see, for instance Sadleir KR et al. Mol Neurodegener. 2015 Jan 7;10:1. doi: 10.1186/1750-1326-10-1.)

Reply: We added features of 5XFAD mice in Introduction (L.36-L.40).

Moreover, the authors introduce the model as Aβ-treated subjects, which is somewhat confusing and may yield some concerns about the methodological approach. This issue turns up also in the Discussion where sentences such as “The consequent neuronal number under Aβ treatment is the sum of surviving neurons and newborn neurons..” are challenging to interpret.

Reply: The confusing sentence “The consequent neuronal number under Aβ treatment is the sum of surviving neurons and newborn neurons.” was deleted.

A more structured introduction of other studies in this direction, apart from the authors’ previous research work, may enrich the overall clinical value of the findings here reported.

Reply: Introduction was revised not sticking only to our previous study. We mentioned as “Although a decline of neurogenesis in the hippocampal dentate gyrus occurs at 12 – 14 weeks of age in 5XFAD mice [7,8], no studies have reported on neurogenesis in placental extracts. Adult neurogenesis is focused in AD field as a possible preventing and therapeutic strategy.” (L.45-L.48)

  1. The authors relied too heavily on p-values and did not provide other critical statistic information that would help elaborate on the clinical meaningfulness of the study. Confidence intervals, effect size coefficient and balancing data interpretation in light of effect size measures over the p-value. Otherwise, the authors should explain why they deem there is no risk of an over-interpretation of the significance of p-values. There is no mention in the statistical section regarding whether comparison tests have been corrected for multiple comparisons to avoid type I error. Finally, it is not clear how whether the authors assessed normality distribution tests before rolling out the comparison and correlation analysis workflow.

Reply: Thank you for very important suggestion about statistics. According your comments, we mentioned detail procedure of statistics (L.159-L.172).

Data are expressed as the mean ± 95% confidence interval (CI). To determine the statistically significant differences, GraphPad Prism 6 (GraphPad Software, San Diego, CA, USA) was used. Since data of Figure 1 passed normality test, parametric tests were performed. Two-tailed paired t-test (Figure 1A), one-way analysis of variance (ANOVA), post hoc Bonferroni's multiple comparisons test (Figure 1B, 1C, 1E), repeated measures two-way ANOVA, post hoc Bonferroni's multiple comparisons test (Figure 1D) were applied. Data in Figures 2, 3 and 4 reveals a mean value and its distribution in similarly treated cells, but not related to individual differences. This kind of data are generally dealt as parametric data. Therefore, one-way ANOVA, post hoc Bonferroni's multiple comparisons test was performed. An effect size (r) was calculated and shown in graphs. The significance level of P value was set at 5%. Criteria of effect sizes of small, medium and large sizes in t-test are 0.2, 0.5 and 0.8, respectively. Criteria of effect sizes of small, medium and large sizes in one-way ANOVA are 0.1, 0.3 and 0.5, re-spectively.

Effects of JBP-F-02 on dendrite genesis, neurogenesis and memory function were evaluated as sufficiently significant by effect size (r) as well as p value.

  1. The authors state:”This study has some limitations as it is unknown which constituents of JBP-F-02 are related to phenotypes, how this stimulates neurogenesis, or whether the mechanisms of neurogenesis and dendrite genesis overlap or are independent.” However, they do not provide any forward-looking argumentation about potential tolls, studies, and solutions to overcome these methodological and conceptual barriers. This standpoint would support replication studies and increase scientific attention towards this study and its potential.

 Reply: Thank you for positive suggestion for better Discussion. We newly mentioned in Discussion about analyzing neurogenesis mechanism and dendrite genesis mechanism as follows with new references (L.313-L.320; L.327-L.332); “Adult neurogenesis linage is pushed forward via regulating several key transcriptional factors such as Pax6, Tbr2, NeuroD and Tbr1 [13]. At least glutamate is contained in human placenta extract [14]. Signaling mechanism of JBP-F-02 for neurogenesis will be investigated by identification of active constituents contributing neurogenesis and re-lated transcriptional factors. The identification of active constituents for neurogenesis in JBP-F-02 is currently under investigation. Concretely, we try to identify active con-stituents which are transferred in the brain after oral administration of JBP-F-02 by LC-MS/MS annotation.” “This study has some limitations as it is unknown which constituents of JBP-F-02 are related to phenotypes, how this stimulates neurogenesis, or whether the mecha-nisms of neurogenesis and dendrite genesis overlap or are independent. As a dendrite genesis mechanism, we focus on the δ-catenin pathway as one of the candidates. δ-catenin localizes to dendrites and dendritic spines in mature neurons [18] and is re-quired for dendritic formation in the cerebral cortex [19].”

Minor points

The author state, “The primary culture was prepared as previously described”, “The dendrite length was evaluated as described previously”,

I suggest providing supplementary material and specifying whether any change to the previous protocol was made or not.

Some expressions, such as ”previous reports” are overused. I suggest improving the language with a wider variety in the wording/phrasing.

Reply: According your suggestion, we deleted unnecessary overusing word “previous” because we already mentioned detail procedures in Method part. Necessary explanation for each experiment is described. For example, more detailed protocol was mentioned for nestin staining (L.151-156).

Round 2

Reviewer 1 Report

The authors answered on all my comments and improved manuscript. However I have some reservations about the sentence “ As a dendrite genesis  mechanism, we focus on the δ-catenin as one of key molecules” (line 328). It seems to be non-grammatical.

Reviewer 2 Report

The authors addressed all the concerns previously raised and provided a significantly improved version of the manuscript. I have no further comments/recommendation to flag.